# Conformational switches control early maturation of the eukaryotic small ribosomal subunit

**Mirjam Hunziker[1][†], Jonas Barandun[1][†][‡]\*, Olga Buzovetsky[1], Caitlin Steckler[2], Henrik Molina[2], Sebastian Klinge[1]\***

[1]Laboratory of Protein and Nucleic Acid Chemistry, The Rockefeller University, New York, United States; [2]Proteomics Resource Center, The Rockefeller University, New York, United States

**\*For correspondence:**
jonas.barandun@umu.se (JB);
klinge@rockefeller.edu (SK)

[†]These authors contributed equally to this work

**Present address:** [‡]The Laboratory for Molecular Infection Medicine Sweden (MIMS), Science for Life Laboratory, Umeå University, Umeå, Sweden

**Competing interests:** The authors declare that no competing interests exist.

**Abstract** Eukaryotic ribosome biogenesis is initiated with the transcription of pre-ribosomal RNA at the 5' external transcribed spacer, which directs the early association of assembly factors but is absent from the mature ribosome. The subsequent co-transcriptional association of ribosome assembly factors with pre-ribosomal RNA results in the formation of the small subunit processome. Here we show that stable rRNA domains of the small ribosomal subunit can independently recruit their own biogenesis factors in vivo. The final assembly and compaction of the small subunit processome requires the presence of the 5' external transcribed spacer RNA and all ribosomal RNA domains. Additionally, our cryo-electron microscopy structure of the earliest nucleolar pre-ribosomal assembly - the 5' external transcribed spacer ribonucleoprotein – provides a mechanism for how conformational changes in multi-protein complexes can be employed to regulate the accessibility of binding sites and therefore define the chronology of maturation events during early stages of ribosome assembly.
DOI: https://doi.org/10.7554/eLife.45185.001

## Introduction

In yeast, the mature ribosome is a 3.2 MDa ribonucleoprotein complex composed of 79 proteins and four ribosomal RNAs (rRNA) assembled into the small and the large subunit. However, throughout its biogenesis over 200 assembly factors are known to sequentially and transiently associate with pre-ribosomal intermediates that mature as they are trafficked from their origin in the nucleolus through the nucleus to their final cytoplasmic destination (*Klinge and Woolford, 2019*). During the past five decades, the identities of most of the proteins and RNAs involved in this pathway were determined. However, the dynamic nature of this process has hindered our ability to understand the chronology of their assembly and the role of conformational changes in this pathway.

Ribosome assembly starts with the RNA polymerase-I driven transcription of the 35S precursor ribosomal RNA (pre-rRNA) in the nucleolus. This primary transcript contains rRNAs of the small ribosomal subunit (SSU, 18S) and the large ribosomal subunit (LSU, 5.8S and 25S) flanked by external transcribed spacers (5' ETS, 3' ETS) and separated by internal transcribed spacers (ITS1, ITS2). Assembly factors are co-transcriptionally recruited and associate with the nascent pre-rRNA, resulting in a chronology of biogenesis that starts at the 5' end, where the 5' ETS and rRNA for the small subunit reside. The first structurally defined particle that assembles co-transcriptionally is therefore the small subunit processome, a giant precursor of the small subunit (*Dragon et al., 2002*).

Initial studies showed that the depletion of early assembly factors resulted in reduced SSU processome formation (*Osheim et al., 2004*). However, these studies were unable to determine to which pre-rRNA sequences these factors were bound. Recent studies using 3' truncated rRNA mimics

provided first insights into the temporal recruitment of ribosome assembly factors to pre-rRNAs, and surprisingly showed that a large number of factors bind to the 5' ETS region, offering the first evidence for a central role of this pre-rRNA sequence (*Chaker-Margot et al., 2015*; *Zhang et al., 2016a*). The particle containing the 5' ETS region and its associated 26 unique polypeptides was termed the 5' ETS particle (*Chaker-Margot et al., 2015*). Factors associated with individual domains of pre-ribosomal RNA have been characterized using biochemical, genetic and structural biology approaches. These studies highlight that the 5' domain is initially chaperoned by U14 snoRNA and assembly factors including Dbp4 and Efg1 (*Soltanieh et al., 2014*; *Soltanieh et al., 2015*; *Choque et al., 2018*; *Shu and Ye, 2018*), the central domain is bound by snR30 and assembly factors including Rok1, Utp23 and Rrp5 (*Morrissey and Tollervey, 1993*; *Lu et al., 2013*; *Wells et al., 2017*; *Lebaron et al., 2013*), whereas several proteins, including Mrd1 and Nop9 have been implicated in binding later regions of the 18S precursor (*Zhang et al., 2016b*; *Segerstolpe et al., 2013*; *Wang and Ye, 2017*). A key insight into small subunit processome formation was the observation that these particles can undergo a structural compaction as shown on Miller spreads (*Osheim et al., 2004*).

Based on genetic data, a hierarchical model was proposed (*Pérez-Fernández et al., 2007*; *Pérez-Fernández et al., 2011*), which was later extended to suggest that the 5' ETS and its associated factors serve as a mold for the assembly of downstream factors and rRNA domains (*Kornprobst et al., 2016*).

Recent structural studies of the fully assembled SSU processome have shown that the 5' ETS and its associated multi-protein complexes UtpA, UtpB and U3 snoRNP form a scaffold for the assembly of individual rRNA domains, thus further highlighting the central role of the 5' ETS (*Sun et al., 2017*; *Barandun et al., 2017*; *Cheng et al., 2017*).

However, since the molecular basis for a hierarchical assembly model of the SSU processome remains elusive, two fundamentally different models can explain the available data:

Early assembly factors could provide a direct and hierarchical support for the recruitment of subsequent proteins and RNAs. Alternatively, individual pre-rRNA sequences could independently recruit their assembly factors and may be mutually dependent on each other for the final SSU processome assembly.

Here we determine the mechanisms underlying the controlled assembly of the SSU processome by using a combination of biochemical, mass-spectrometry and cryo-electron microscopy (cryo-EM) analyses of nucleolar particles preceding the SSU processome. We show that the 5' ETS and stable rRNA domains are functionally independent in their ability to recruit their own assembly factors, and the strict order of events is enforced through molecular switches within the 5' ETS ribonucleoprotein (RNP) that require the presence of all rRNA domains for the final SSU processome to be formed.

## Results

### Stable 18S rRNA domains independently recruit their assembly factors

To determine the order in which ribosome assembly factors associate with pre-rRNA, previous systems employed 3' end truncations of rRNA mimics to emulate continuing transcription (*Chaker-Margot et al., 2015*; *Zhang et al., 2016a*). As a result of this design, these systems cannot distinguish between factors associating with longer pre-rRNA constructs due to independent or hierarchical binding. To address if each rRNA domain can independently recruit its own assembly factors, we modified our previous in vivo system (*Chaker-Margot et al., 2015*) (*Figure 1a*, *Figure 1—figure supplement 1*). Each MS2 aptamer-tagged pre-rRNA domain was expressed in a strain containing a tagged ribosome assembly factor that was known to associate with it (Utp10, Esf1, Kri1 and Mrd1 for 5' ETS, 5' domain, central domain and 3' major domain, respectively). Tandem affinity purifications were performed using first the tagged RNA via a co-expressed MS2-3C-GFP fusion protein followed by the streptavidin-binding-peptide-tagged ribosome assembly factor (*Figure 1—figure supplement 1*). All rRNA mimics, except for the 3' major domain in isolation, were expressed and associated with ribosome assembly factors as shown by Northern blotting and SDS-PAGE analysis of the purified ribonucleoprotein complexes respectively (*Figure 4—figure supplement 1*). While the control pre-rRNA construct containing the 5' ETS and all rRNA domains was able to capture all SSU processome factors that were previously observed in the fully assembled yeast SSU processome

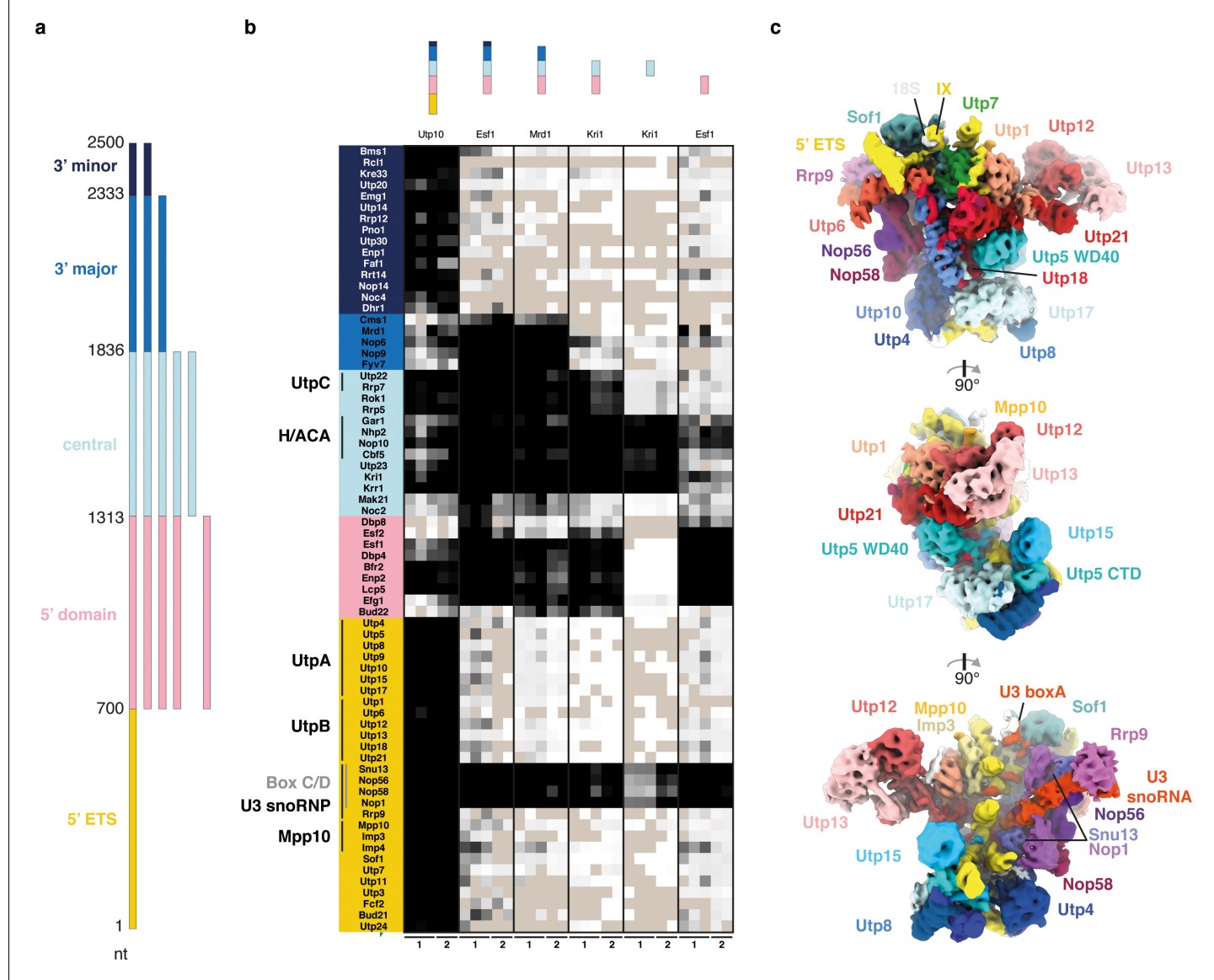

**Figure 1.** The 18S rRNA domains recruit assembly factors independent of the 5' ETS. (a) Schematics of rRNA mimics with color-coded rRNA domains. (b) Bait-normalized (MS2-protein) iBAQ based heat-map (proteins not detected in light brown, low abundance to high abundance in gradient from white to black) of ribosome biogenesis factors co-purified with pre-rRNA constructs shown in (a). Each biological replicate (n = 2) is labeled at the bottom and all technical replicates (n = 3, n = 2) are shown. (c) Three 90° related views of the cryo-EM structure of the 5' ETS RNP lowpass-filtered to 5 Å with density colored according to subunits. Subunits of UtpA (shades of blue), UtpB (shades of red) and U3 snoRNP (shades of purple with U3 snoRNA in red) are shown.

DOI: https://doi.org/10.7554/eLife.45185.002

The following figure supplements are available for figure 1:

**Figure supplement 1.** Expression and purification system used to isolate RNPs containing defined rRNA mimics.
DOI: https://doi.org/10.7554/eLife.45185.003

**Figure supplement 2.** Cryo-EM analyses of the 5' ETS RNP in different pre-ribosomal intermediates.
DOI: https://doi.org/10.7554/eLife.45185.004

**Figure supplement 3.** Cryo-EM data processing strategy, overall and local resolution estimation of the 5' ETS RNP.
DOI: https://doi.org/10.7554/eLife.45185.005

(*Barandun et al., 2017*), surprising patterns emerged for the other isolated pre-rRNA domains (*Figure 1b*). Each of the stable rRNA domains directly recruited the factors stably associated with that domain in the context of the complete SSU processome, as well as factors that are only transiently bound and dissociate upon SSU processome formation (*Figure 1b*). While the 5′ domain and central domain could be stably expressed in isolation, the 3′ major domain was unstable when expressed individually, which prevented a direct analysis of its bound factors (*Figure 1b*, *Figure 4—figure supplement 1a,b*).

Strikingly, proteins that were previously found to require the 3′ minor domain of the 18S rRNA for their association with the SSU processome (*Chaker-Margot et al., 2015*; *Zhang et al., 2016a*) also require the 5′ ETS (*Figure 1b*). Combined, these data show that stable individual rRNA domains can directly recruit their associated factors and that the 5′ ETS and the 3′ region of the 18S rRNA are both required for final SSU processome formation. The critical role of the 5′ ETS particle for SSU processome formation prompted us to examine its structure by cryo-EM.

## Structure of the 5′ ETS RNP

We determined cryo-EM structures of the 5′ external transcribed spacer in the context of several nucleolar precursors containing either the 5′ ETS alone (17 Å), the 5′ ETS and 5′ domain (9.5 Å) or a pre-rRNA mimic covering the 5′ ETS, 5′ domain and the central domain (4.3 Å) (*Figure 1c*, *Figure 1—figure supplements 2–3*, *Figure 4—figure supplements 2–3*, *Supplementary file 2*). Although individual rRNA domains are flexible and functionally independent modules, we observed the 5′ ETS RNP as the unifying and stable structural entity in each of these reconstructions.

Multi-protein complexes UtpA and UtpB as well as the U3 snoRNP constitute the major components of this assembly. Additional proteins such as Utp7, Sof1, and Imp3 with its associated segment of Mpp10 are clearly visible (*Figure 1c*). Consistent with their involvement in bridging distant sites within the maturing SSU processome (*Barandun et al., 2018*), 5′ ETS proteins Imp4, Sas10, Utp11, Bud21 and large parts of Mpp10 are bound, as shown by mass spectrometry data for each particle (*Supplementary file 2*), but not yet completely ordered in these reconstructions. In the earliest form of the 5′ ETS particle, which lacks ribosomal RNA, parts of UtpB (Utp6), Sof1 and the 5′ ETS (helix VII-IX) are also disordered (*Figure 1—figure supplement 2a,d*). In later forms of the 5′ ETS particle, Utp6, Sof1 and 5′ ETS helices VII-IX as well as an RNA duplex containing U3 snoRNA box A become ordered (*Figure 1c* and *Figure 1—figure supplement 2b–d*). Strikingly, the 5′ ETS RNP adopts a structure with significant differences with respect to the SSU processome (*Figure 2a*).

## Conformational switches in the 5′ ETS RNP coordinate the timing of maturation factor recruitment

Conformational switches within the 5′ ETS RNP contribute to the regulated formation of the SSU processome by inhibiting binding of late assembly factors that are associated with the compaction of the SSU processome upon completed transcription of the 18S rRNA (*Chaker-Margot et al., 2015*; *Zhang et al., 2016a*; *Barandun et al., 2018*). Preventing the premature binding of these late factors likely constitutes an important regulatory step, which is catalyzed by UtpB, UtpA and parts of the pre-rRNA.

Compared with the SSU processome, the most dramatic conformational changes occur within the UtpB complex (*Figure 2b–c* and *Figure 2—video 1*). This complex adopts a conformation that reconfigures its tetramerization module as well as the tandem β-propellers of Utp12 and Utp13 into a compacted state (*Figures 2* and *3a,b*). As a result of this large conformational change, several binding sites for other assembly factors are disrupted, such as the bipartite binding site of the late factor Pno1, the binding sites for the C-terminal region of Mpp10, the Rcl1•Bms1 complex, several ribosomal proteins, and the 18S pre-rRNA (*Figure 3c,d*). Similarly, the binding sites for Nop14 and Noc4 are formed on the UtpA platform (Utp15), which is initially distorted in the 5′ ETS RNP (*Figure 3e,f*).

In contrast to previously characterized SSU processome particles, which were cut at site A0 (*Barandun et al., 2017*), the incomplete rRNA mimics used in our study did not exhibit significant cleavage at either A0 or A1 (*Figure 4—figure supplements 2c–e* and *3*). Continuous density extending from helices VII and VIII of the 5′ ETS is for the first time observed between Sof1 and Utp7. This density corresponds to the base of helix IX of the 5′ ETS and contains the A0 cleavage

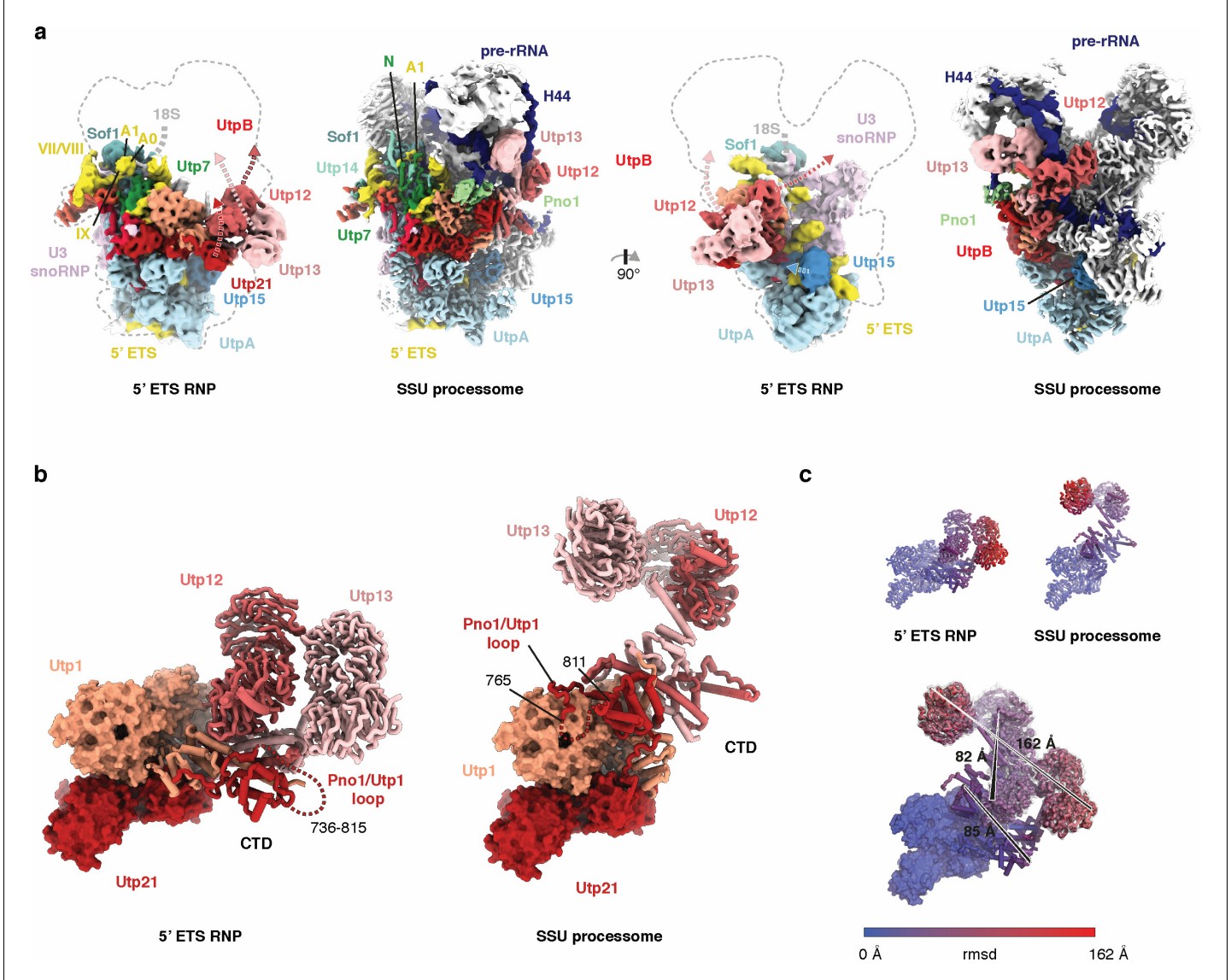

**Figure 2.** Conformational differences of UtpB in the context of the 5' ETS RNP and the SSU processome. (a) Two views of the 5 Å lowpass-filtered cryo-EM density of the 5' ETS RNP (left) and the SSU processome (*Barandun et al., 2017*) (right) (EMD-8859). Subunits and dashed arrows indicating conformational changes upon SSU processome formation are color-coded. (b) UtpB conformations with its subunits colored in shades of red as in (a) in the context of 5' ETS RNP and the SSU processome. (c) UtpB conformations in the context of the ETS RNP, the SSU processome and both superimposed, colored according to root mean square deviation (rmsd) between the two conformations. The largest conformational differences are indicated with a black line and the distances between the two positions are labeled.

DOI: https://doi.org/10.7554/eLife.45185.006

The following video is available for figure 2:

**Figure 2—video 1.** A conformational change in UtpB creates binding sites for late assembly factors.

DOI: https://doi.org/10.7554/eLife.45185.007

site (*Figure 1—figure supplement 2d*). In the 5' ETS RNP, an RNA linker and helix IX sterically prevent the association of the late binding factor Utp14, which binds in the same region in the SSU processome after A0 cleavage or RNA remodeling (*Figure 4a–d*).

## Compaction as SSU processome quality control checkpoint

Within the SSU processome, U3 snoRNA acts as a central organizer, which base-pairs with two segments of both the 5' ETS (5' and 3' hinges) and the 18S pre-rRNA (box A and box A') (*Figures 1c*

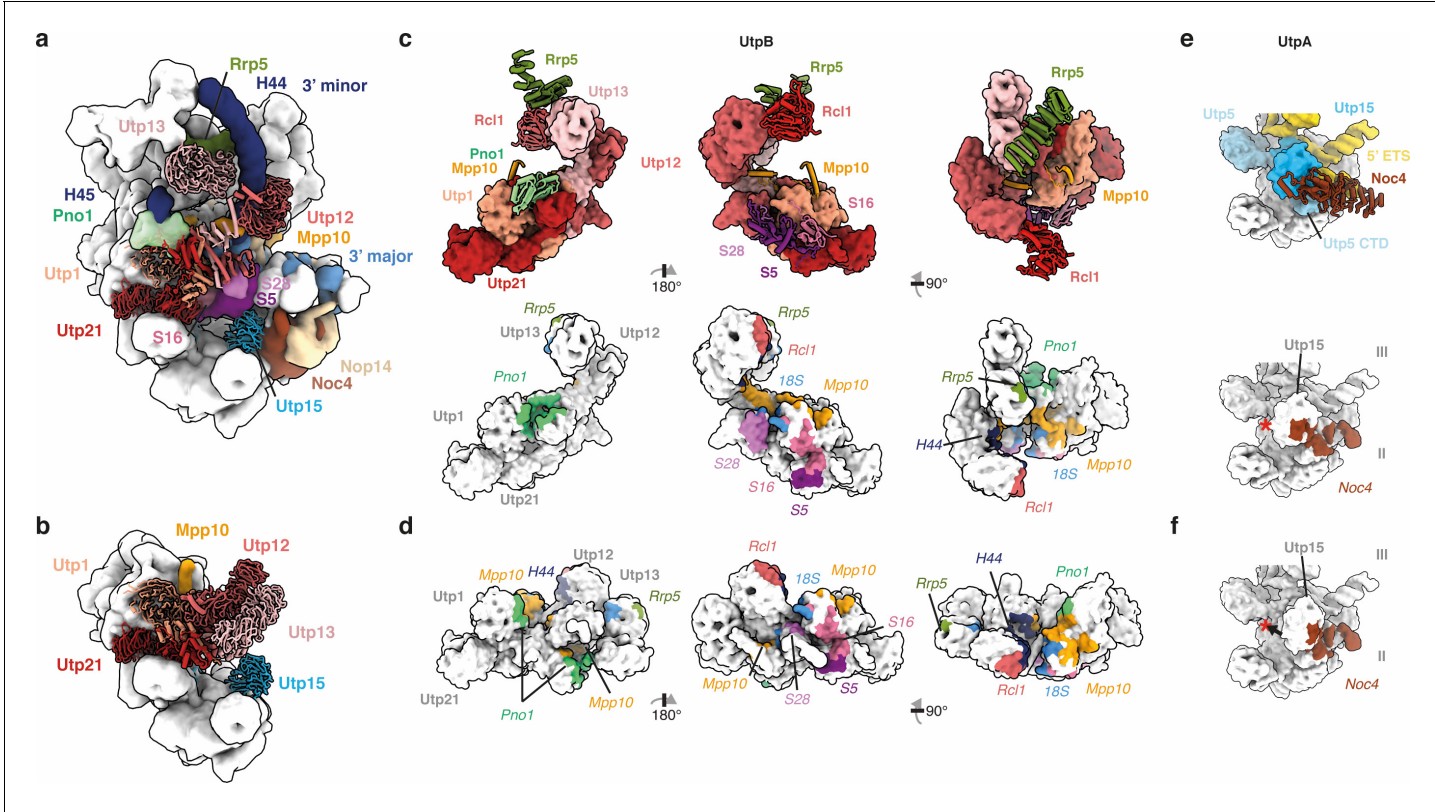

**Figure 3.** Conformational switches in the 5' ETS RNP coordinate the timing of maturation factor recruitment. (**a, b**) UtpB (shades of red) in the SSU processome (**a**) and the 5' ETS RNP (**b**). Interaction partners of UtpB and Utp15 (UtpA) in the SSU processome are shown and labeled. (**c, d**) UtpB in its SSU processome conformation (**c**) and 5' ETS RNP conformation (**d**). UtpB, colored as in (**a**) is shown as surface with direct interaction partners visualized as cartoon (c, top panel). Protein and RNA interaction interfaces (*Krissinel and Henrick, 2007*) of UtpB within the SSU processome are color-coded according to (**a**) onto the surface of UtpB in its SSU processome conformation (c, lower panel) or in its 5' ETS RNP conformation (**d**). (**e, f**) Utp15 (UtpA) in context of the SSU processome (**e**) and the 5' ETS RNP (**f**). Movement of Utp15 is indicated with an arrow and a red star. (**c–f**) Interaction interfaces are colored as the interacting factor in (**a**) and labeled in italics.

DOI: https://doi.org/10.7554/eLife.45185.008

The following figure supplement is available for figure 3:

**Figure supplement 1.** Peptides associated with the 5' ETS RNP and their subsequent interaction interfaces.

DOI: https://doi.org/10.7554/eLife.45185.009

and *4a–d*), thereby defining the positioning of rRNA domains within the SSU processome (*Sun et al., 2017*; *Barandun et al., 2017*; *Barandun et al., 2018*; *Chaker-Margot et al., 2017*). While the 5' and 3' major domains are encapsulated by assembly factors, the central domain is more dynamic and bound by fewer proteins including the UtpC complex, Rrp5 and Kri1 (*Barandun et al., 2017*). These observations, together with our mass spectrometry analysis (*Figure 1b*) prompted us to probe SSU processome quality control by testing if the base-pairing between U3 snoRNA and pre-18S in the absence of the central domain would be sufficient for SSU processome formation (*Figure 5*). The comparison of proteins that copurify with two pre-rRNA segments either containing or lacking the central domain illustrated the high fidelity with which SSU processome assembly is controlled. While the truncated pre-rRNA containing all 18S subdomains was able to form SSU processomes as highlighted by the presence of late factors including Bms1, Utp20 and Kre33, the truncated pre-rRNA lacking the central domain associated with factors that bind the 5' domain (Dbp8, Esf2, Dbp4, Esf1 and Efg1) and 3' major domain (Nop6, Cms1) transiently during early SSU processome formation but did not associate with either central domain or late binding factors. Surprisingly, the nuclear exosome was recruited to the formed SSU processome rather than the construct lacking the central domain, suggesting a function in pre-rRNA processing. These data show

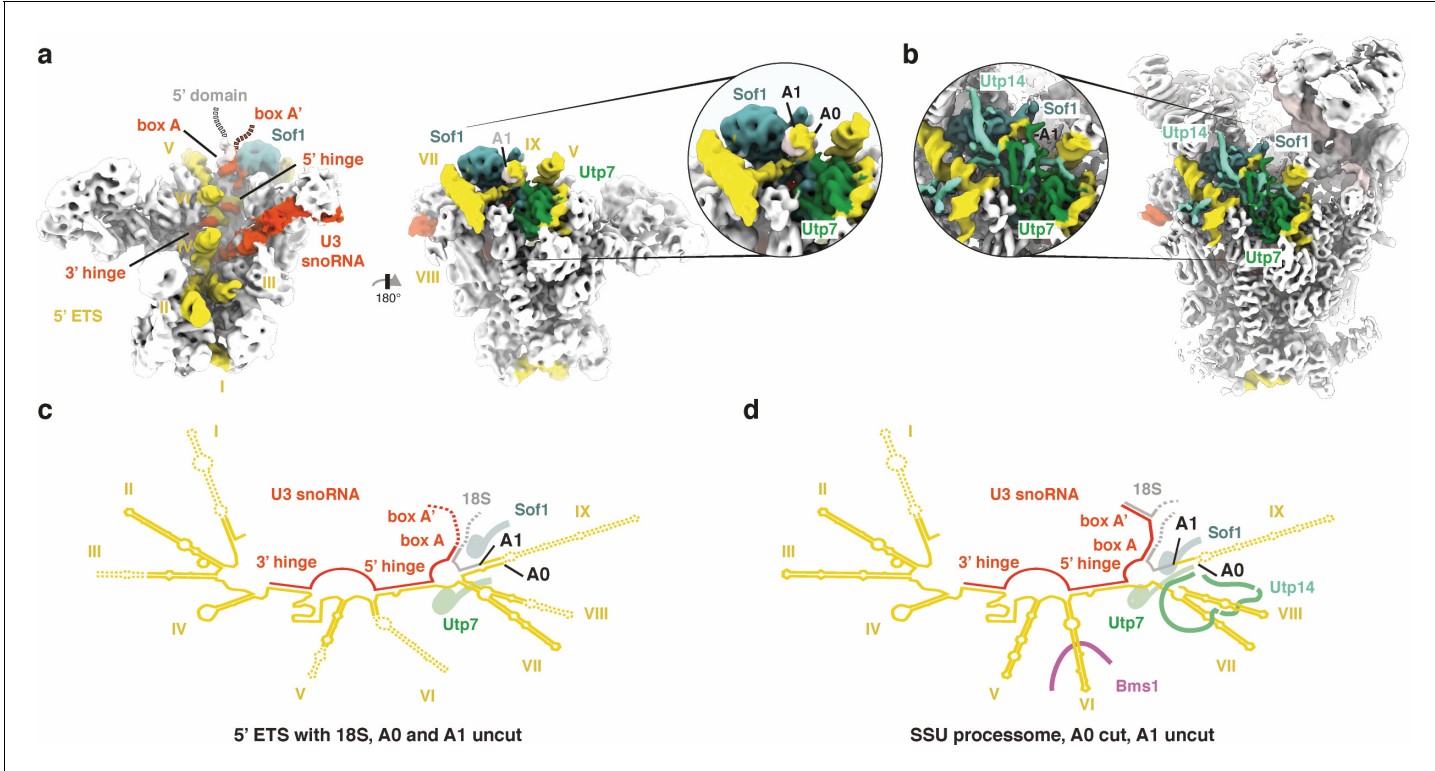

**Figure 4.** A0 cleavage creates a binding site for Utp14. (a, b) 5' ETS, U3 snoRNA and sites A0/A1 in the cryo-EM volume of (a) the 5' ETS RNP and (b) the SSU processome (EMD-8859). Insets highlight the approximate positions of sites A0/A1. (c, d) Secondary structure of the 5' ETS (yellow), the U3 snoRNA (red) and the 18S rRNA (gray) as observed in (a) and (b). Disordered regions are indicated by dashed lines.

DOI: https://doi.org/10.7554/eLife.45185.010

The following figure supplements are available for figure 4:

**Figure supplement 1.** Purification of 18S-rRNA domain-containing particles used for mass spectrometry analysis.
DOI: https://doi.org/10.7554/eLife.45185.011

**Figure supplement 2.** Biochemical characterization of small subunit assembly stages preceding SSU processome formation.
DOI: https://doi.org/10.7554/eLife.45185.012

**Figure supplement 3.** Analysis of degradation products in early assembly intermediates.
DOI: https://doi.org/10.7554/eLife.45185.013

that while each rRNA domain can act as an independent functional module, the presence of all rRNA domains is required for SSU processome formation (*Figure 6*).

## Discussion

### A new model for small subunit processome assembly

The architecture of the 5' ETS RNP together with analyses of defined in vivo assembled pre-rRNA particles have shed light on the mechanisms of quality control during the formation of the SSU processome. While we cannot categorically rule out the possibility that the particles isolated in this study are off-pathway intermediates, the following observations suggest that this is not the case. Our previous work demonstrated that pre-rRNA mimics containing MS2 loops that are transcribed by RNA polymerase II can serve as the sole source of ribosomal RNA (*Chaker-Margot et al., 2015*). Furthermore, in good agreement with prior genetic and biochemical data, we detected all expected ribosome assembly factors that were associated with the studied pre-rRNA mimics. Lastly, the specific A0 cleavage observed upon SSU processome formation indicates that the studied particles undergo specific quality control as expected for native particles.

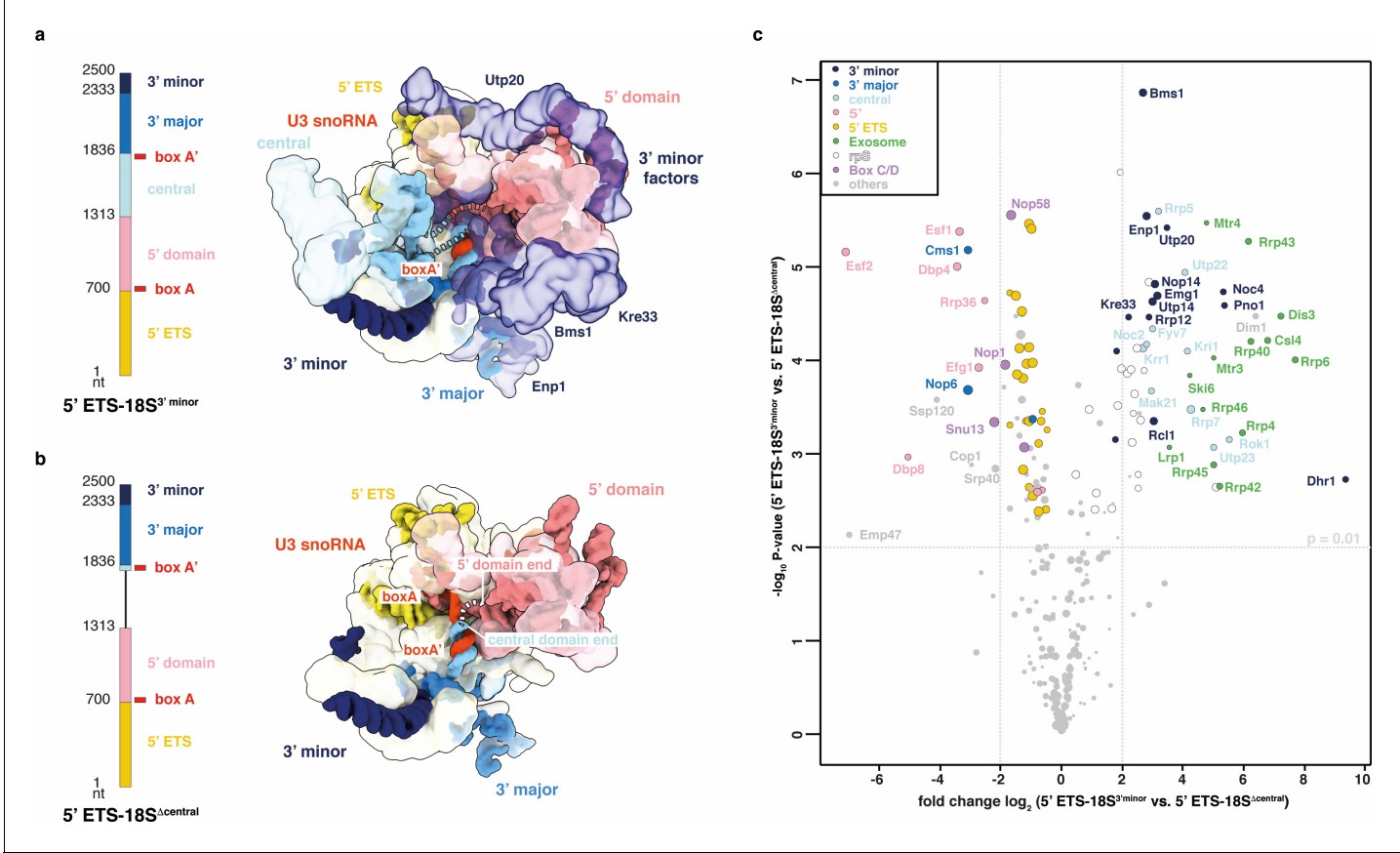

**Figure 5.** Compaction as SSU processosome quality control checkpoint. (**a**, **b**) rRNA transcripts used in (**c**) with individual pre-rRNA domain boundaries and U3 snoRNA base-pairing sites (box A, A'). Top view of the SSU processosome (PDB 5WLC) (**a**) and a theoretical model of the same particle without the central domain (**b**). pre-rRNA domains and the proteins (transparent) they recruit are color-coded. Dotted lines indicate connections between pre-rRNA domains. (**c**) Volcano plot showing the label-free quantification (LFQ) comparison for proteins identified in 5' ETS-18S$^{3' \, minor}$ vs. 5' ETS-18S$^{\Delta \, central}$. Log-transformed fold changes (x-axis) and p-values (y-axis). Proteins are color-coded and their relative abundance indicated by sphere size.

DOI: https://doi.org/10.7554/eLife.45185.014

Contrary to prior models that suggested a hierarchical assembly of SSU processosome factors (*Pérez-Fernández et al., 2007*; *Pérez-Fernández et al., 2011*) or molding of the pre-rRNA (*Kornprobst et al., 2016*), our data provides evidence for a new model of SSU processosome maturation. In this model, the early steps in eukaryotic ribosome assembly are governed by the initial functional independence of the 5' ETS and stable rRNA domains, which provides sufficient flexibility for parallel maturation and can explain the high efficiency of ribosome synthesis (*Figures 1*, *5* and *6*). The structure of the 5' ETS RNP has further illustrated that some assembly factors (such as Mpp10, Sas10, Utp11) can bind to the earliest precursor with a high degree of flexibility before subsequent folding occurs once all rRNA domains have been assembled within the SSU processosome (*Figure 6a–c*, *Figure 3—figure supplement 1*). A particularly important module of the 5' ETS RNP is the UtpB complex, which acts as a molecular switch and molecular sensor of pre-rRNA transcription. Disabled binding sites within UtpB prevent the premature association of later factors (*Figure 6a–d*). Once transcription reaches the end of the 18S and all rRNA domains have matured, UtpB changes its conformation and allows for SSU processosome formation (*Figure 6e*). These emerging principles have general implications for the highly regulated assembly of many large eukaryotic protein and RNA-protein complexes (*Mimaki et al., 2012*; *Wild and Cramer, 2012*; *Fica and Nagai, 2017*) that are subject to extensive control mechanisms.

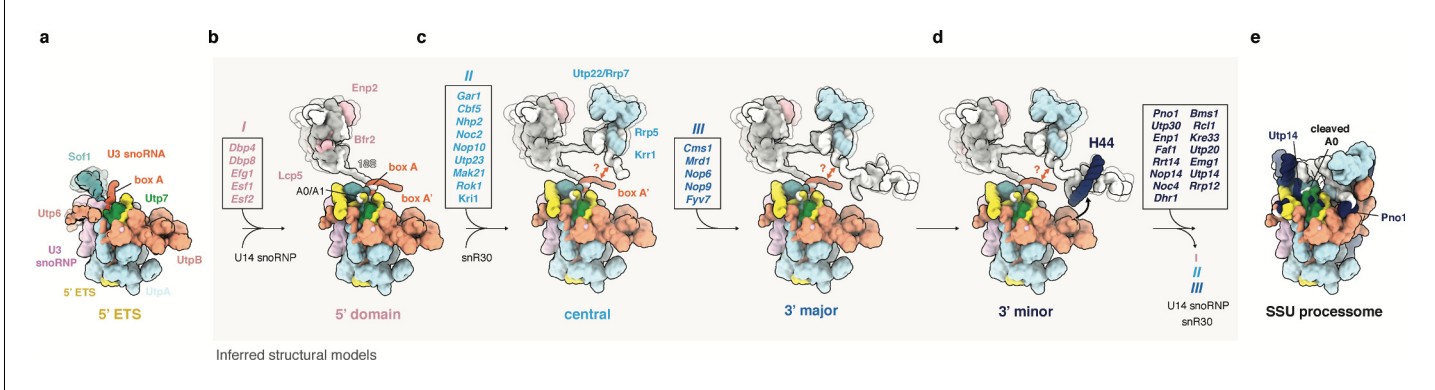

**Figure 6.** Model of SSU processome formation. (**a**) The 5' ETS recruits UtpA (blue), UtpB (light red), U3 snoRNP (pink, RNA red) and additional factors such as Utp7 (green) and Sof1 (blue). Parts of Utp7 and Sof1 are flexible in the absence of the 18S rRNA and UtpB adopts a retracted conformation as shown by the presented cryo-EM structure of the 5' ETS RNP. (**b–d**) The 5' ETS RNP forms a separate module during the transcription and independent maturation of the 18S rRNA domains (5', central, 3'major, 3'minor). Individual 18S rRNA domains can independently recruit assembly factors. The inferred structures of subsequent assembly intermediates (**b–d**) are based on the structure of the 5' ETS RNP and individual rRNA domains and associated factors as seen in the fully assembled SSU processome (**e**, PDB 5WLC). Assembly factors transiently associated with a particular assembly stage are boxed and labeled (*I, II, III*). Factors in these boxes (*I, II, III*) together with U14, snR30 and others leave the maturing particle before SSU processome formation as indicated by an arrow in (**d**). (**d**) UtpB acts as a sensor for the 3' end of the 18S rRNA by recognizing elements in the 3' minor domain (H44, dark blue). A conformational change in UtpB and the recruitment of 3' minor factors (dark-blue, boxed) lead to the formation of the SSU processome. (**e**) In the fully assembled SSU processome (PDB 5WLC) the 5' ETS is cleaved at site A0. Pno1 is stabilized by the interaction with the rotated UtpB CTD tetramer and Utp14 occupies a binding site obstructed by a 5' ETS RNA linker in the 5' ETS RNP.

DOI: https://doi.org/10.7554/eLife.45185.015

# Materials and methods

## Cloning of MS2-tagged 18S-rRNA domains and the MS2-3C-GFP construct

Defined segments of the rDNA locus of *S. cerevisiae* strain BY4741 were cloned into a derivative of the pESC_URA vector (Agilent Technologies). Primers used to amplify subsections of the rDNA locus are listed in *Supplementary file 3*. The rDNA domains were tagged with five MS2-aptamer stem loops at their 3′ ends and cloned downstream of a *gal1* promoter and upstream of a CYC terminator (*Supplementary file 3*). A modified MS2-coat protein (*LeCuyer et al., 1995*) fused to an N-terminal nuclear localization signal (NLS) as well as a hemagglutinin (HA) tag and a C-terminal 3C-protease-cleavable GFP (NLS-HA-MS2-3C-GFP) was cloned into a modified pESC plasmid suitable for genome integration in yeast (B079: *gal10* promoter, G418 resistance) (*Supplementary file 3*).

## Yeast strains

B079 was linearized and integrated into the *pep4* locus of *S. cerevisiae* strain BY4741 by homologous recombination to yield Y180, which was used to create all subsequent yeast strains harboring C-terminal streptavidin-binding-peptide (SBP) tags on endogenous ribosome biogenesis factors (Utp10, Esf1, Kri1 and Mrd1) (*Supplementary file 3*). Homologous recombination-based integration events of C-terminal SBP tags were selected for with Hygromycin or Nourseothricin resistance cassettes. Yeast transformation and genetic manipulations were performed according to established procedures.

## Expression of pre-rRNA mimics

Yeast strains harboring a galactose-inducible NLS-HA-MS2-3C-GFP in the *pep4* locus and a streptavidin binding peptide (SBP)-tagged ribosome assembly factor (*Figure 1—figure supplement 1*) were transformed with plasmids carrying a URA marker and rDNA constructs (*Supplementary file 3*): strain Y186 (Utp10-SBP) was used for all experiments with plasmids containing the 5' ETS sequence (B117, B221 to B224 and B514), Y367 (Esf1-SBP) for the 5' domain containing plasmid

(B506) and the 18S rRNA plasmid (B504), Y372 (Kri1-SBP) for the central domain plasmids (B502, B503) and Y374 (Mrd1-SBP) for the 3' major domain containing plasmid (B495, B496).

Transformed cells were grown on -URA synthetic dropout media plates supplemented with 2% glucose (w/v) and appropriate antibiotics (Supplementary file 3) for 2 days at 30℃. Selected colonies were picked and grown in pre-cultures (100 ml) of -URA synthetic dropout media supplemented with 2% raffinose (w/v) and selective antibiotics to an optical density (OD) of 2 at 600 nm. Large-scale cultures were inoculated with the pre-cultures and grown at 30℃ for 16 hr to an OD of 4.5–7 in the presence of 2% galactose. Yeast cells were pelleted, washed twice with ddH₂O and once with ddH₂O supplemented with protease inhibitors (PMSF, Pepstatin A, E-64). The cells were flash frozen in liquid N₂ and subsequently cryo-ground using a planetary ball mill (Retsch PM100).

## Purification of overexpressed pre-rRNA mimic-containing RNPs

10–40 grams of cryo-ground yeast powder were resuspended in binding buffer (50 mM Tris-HCl, pH 7.5 (RT), 150 mM NaCl, 5 mM MgCl₂, 5% glycerol, 0.1% NP40) supplemented with protease inhibitors (PMSF, Pepstatin A, E-64). Insoluble fractions were pelleted for 20 min at 40'000 x g, 4℃, and the supernatant was incubated with anti-GFP nanobody coupled sepharose (Chromotek) for 3 to 4 hr at 4℃. Pre-rRNA mimics and their associated proteins were eluted by 3C-protease cleavage at 4℃ for 1 hr. Subsequently, eluted RNPs were applied to Streptavidin-coupled sepharose resin for 1 hr at 4℃, washed four times with wash buffer (50 mM Tris-HCl, pH 7.5 (RT), 150 mM NaCl, 5 mM MgCl₂) and released from beads by incubation in 50 mM Tris-HCl, pH 7.5 (RT), 150 mM NaCl, 5 mM MgCl₂, 5 mM D-biotin for 30 min at 4℃. Eluted samples were either directly used for RNA extraction, mass-spectrometry analysis and negative stain electron microscopy sample preparation, or supplemented with 5 mM putrescine, 1 mM spermidine and 0.03% Triton X-100 for subsequent cryo-electron microscopy studies.

## Negative-stain electron microscopy analysis

3.5 µl of pre-ribosomal particles purified from Y186 transformed with B117 (5' ETS), B221 (5' ETS-18S[5'-domain]), B222 (5' ETS-18S[central-domain]), B223 (5' ETS-18S[3'major-domain]) or B224 (5' ETS-18S[3'minor domain]) were applied on glow-discharged carbon coated grids (EMS, CF200-Cu). Subsequently, grids were washed three times with ddH₂O, twice with 2% (w/v) 0.2 µm-filtered uranyl acetate and air dried. Micrographs were acquired on a Philips CM10 operated at an acceleration voltage of 100 kV equipped with a XR16-ActiveVu (AMT) camera at a nominal magnification of 39,000 and a calibrated pixel size of 3.4 Å at the specimen level.

## Cryo-EM grid preparation

Samples in elution buffer (50 mM Tris-HCl, pH 7.5 (RT), 150 mM NaCl, 5 mM MglCl₂, 5 mM D-biotin) with absorbances of 0.35 mAU (5' ETS, B117), 0.65 mAU (5' ETS-18S[5'-domain], B221), 0.8 mAU (5' ETS-18S[central-domain], B222) at 260 nm (Nanodrop 2000, Thermo Scientific) were supplemented with 5 mM putrescine, 1 mM spermidine and 0.03% Triton X-100. 3.5 µl of sample was applied on glow-discharged lacey-carbon grids containing a thin carbon film (TED PELLA, Inc, Prod. No. 01824). Following a 15 s sample incubation period at close to 100% humidity, grids were blotted for 1.5–2.5 s with a blotting force of 0 and flash frozen in liquid ethane using a Vitrobot Mark IV (Thermo Fisher Scientific).

## Cryo-EM data collection and processing

Cryo-EM data collection was performed either on a Talos Arctica or Titan Krios (Thermo Fisher Scientific) operated at 200 kV or 300 kV respectively, both mounted with a K2 Summit detector (Gatan, Inc). SerialEM (Mastronarde, 2005) was used for automated data collection. Datasets of the 5' ETS (1199 micrographs, 1.2 Å pixel size, eight electrons per pixel and second) and the 5' domain particle (697 micrographs, 1.9 Å pixel size, 15 electrons per pixel and second) were collected on a Talos Arctica and processed using RELION-2 (Kimanius et al., 2016). As a starting model, a CryoSPARC (Punjani et al., 2017) generated initial model obtained from the 5' domain data set was used. While the 5' ETS particle could not be refined to a high resolution (~17 Å) due to heterogeneity in the sample, we were able to obtain a ~ 10 Å reconstruction of the 5' domain particle with similar overall

structure but better resolved density for the A1 binding site of Sof1 and Utp7 (*Figure 1—figure supplement 2, a* to c).

The central-domain particle dataset was acquired on a Titan Krios: 2750 movies with 32 frames over an exposure time of 8 s at a dose rate of 10 electrons per pixel and second (total dose of 31.25 $e^-/Å^2$) over a defocus range of 1–3.5 µm at 1.6 Å pixel size (super-resolution pixel size 0.8 Å). Motion correction within RELION-3 (*Zivanov et al., 2018*) was used for gain normalization, beam-induced motion correction and dose-weighting. The contrast transfer function was estimated with CTFFIND-4.1.5 (*Rohou and Grigorieff, 2015*). Removal of micrographs with bad CTF fits resulted in a total of 2'592 micrographs used for reference-free particle picking with gautomatch (http://www.mrc-lmb.cam.ac.uk/kzhang/) yielding 275'080 particles (*Figure 1—figure supplement 3c*). Particles were extracted with a box size of 360 pixel (576 Å) and subjected to 3D classification in RELION-3 with 3, 4, 5 and 7 classes and the 5' domain structure low-pass filtered to 60 Å as initial model (*Figure 1—figure supplement 3c*). From the different 3D classification runs, top classes were selected and combined to result in 180'274 unique particles. These were used for focused refinement and postprocessing resulting in a map with an overall resolution of 4.2 Å but less well resolved peripheral regions (*Figure 1—figure supplement 3c*). A subsequent 3D classification with seven classes resulted in one class (52'629 particles) with improved density for the more peripheral regions (UtpB, U3 snoRNP). This class was refined to a final resolution of 4.3 Å and deposited (EMD-0441) (*Figure 1—figure supplement 3d and e*, and *Table 1*).

## Model building

The structure of the 5' ETS RNP moiety of the small subunit processome (PDB 5WLC) (*Barandun et al., 2017*) was used as initial coordinates for model building in the 4.3 Å map. The entire starting coordinates of the 5' ETS RNP part of the SSU processome were docked as one entity into the density using UCSF Chimera (*Pettersen et al., 2004*). All subunits were then individually rigid body fitted and trimmed in COOT. Major differences were observed in the six-subunit complex UtpB, which required rigid body docking of individual subunit domains (C-terminal tetramerization domains and tandem β-propellers). Additional helical density next to Utp12 could not be unambiguously assigned and therefore a poly-Alanine helix was placed. The structure was refined using phenix.real_space_refine (*Adams et al., 2010*) with secondary structure restraints obtained from the model and (PDB 5WLC). Removing of side-chains resulted in a poly-alanine model with residue information (PDB 6ND4). Data collection and processing information as well as model statistics can be found in *Table 1*. Molecular graphics and analyses were performed with UCSF ChimeraX (*Goddard et al., 2018*) and PDBePISA (*Krissinel and Henrick, 2007*).

## RNA extraction and northern blots

Total cellular RNA was extracted from 0.2 gram of frozen yeast cells after lysis by bead beating in 1 ml Trizol (Ambion). To isolate RNA from purified pre-ribosomal particles, 0.5 ml Trizol (Ambion) was added to the final D-biotin elutions and the extraction was performed according to the manufacturer's instructions. For the analysis of pre-rRNA processing states by Northern blotting, 3 µg total cellular RNA or ~100 ng of RNA extracted from purified RNPs, were loaded in each lane of a 1.2% agarose formaldehyde-formamide gel and separated at 75V for 2.5 hr. After running, the separated RNA was transferred onto a cationized nylon membrane (Zeta-Probe GT, Bio-Rad) using downward capillary transfer and cross-linked to the membrane for Northern blot analysis by UV irradiation at 254 nm with a total exposure of 120 millijoules/cm$^2$ in a UV Stratalinker 2400 (Stratagene).

Prior to the addition of γ−32P-end-labeled DNA oligo nucleotide probes (*Supplementary file 3*), cross-linked membranes were incubated with hybridization buffer (750 mM NaCl, 75 mM trisodium citrate, 1% (w/v) SDS, 10% (w/v) dextran sulfate, 25% (v/v) formamide) for 30 min at 65℃. Labeled hybridization probes were incubated with the membrane first at 65℃ for 1 hr and then at 37–45℃ overnight. Blotted membranes were washed once with wash buffer 1 (300 mM NaCl, 30 mM trisodium citrate, 1% (w/v) SDS) and once with wash buffer 2 (30 mM NaCl, 3 mM trisodium citrate, 1% (w/v) SDS) for 30 min each at 45℃, before radioactive signal was read out by exposure of the washed membranes to a storage phosphor screen, which was subsequently scanned with a Typhoon 9400 variable-mode imager (GE Healthcare).

**Table 1.** Cryo-EM data collection, refinement and validation statistics.

| | Structure of the 5' ETS RNP PDB 6ND4 EMD-0441 |
|---|---|
| **Data collection and processing** | |
| Voltage (kV) | 300 |
| Pixel size (Å) | 1.6 |
| Electron exposure (e- / Å$^2$) | 31.25 |
| Defocus range (um) | 1–3.5 |
| Symmetry imposed | C1 |
| Initial particle images | 275'080 |
| Final particle images | 52'629 |
| Resolution (Å) | 4.3 |
| FSC threshold | 0.143 |
| Map sharpening B-Factor (Å$^2$) | −64.77 |
| | |
| **Refinement** | |
| Initial model used | 5WLC |
| Model composition | |
| Non hydrogen atoms | 64,019 |
| Protein residues | 10309 |
| RNA bases | 569 |
| Ligands | 0 |
| R.m.s. deviations | |
| Bond length (Å) | 0.01 |
| Angles (°) | 1.25 |
| Validation | |
| MolProbity score | 1.71 |
| Clashscore | 5.26 |
| Poor rotamers (%) | 0.0 |
| Good sugar puckers (%) | 98.7 |
| Ramachandran | |
| Favored (%) | 93.39 |
| Allowed (%) | 6.56 |
| Outliers (%) | 0.05 |

DOI: https://doi.org/10.7554/eLife.45185.016

## Mass spectrometry sample processing and data analysis

Purified RNP samples were dried and dissolved in 8 M urea/0.1 M ammonium bicarbonate/10 mM DTT. After reduction, cysteines were alkylated in 30 mM iodoacetamide. Proteins were digested with LysC (LysC, Endoproteinase LysC, Wako Chemicals) in less than 4 M urea followed by trypsination (Trypsin Gold, Promega) in less than 2 M urea. Digestions were halted by adding TFA and digests were desalted (*Rappsilber et al., 2007*) and analyzed by reversed phase nano-LC-MS/MS using a Fusion Lumos (Thermo Scientific) operated in high/high mode.

Data were quantified and searched against the *S. cerevisiae* Uniprot protein database (October 2018) concatenated with the MS2-protein sequence and common contaminations. For the search and quantitation, MaxQuant v. 1.6.0.13 (*Cox et al., 2014*) was used. Oxidation of methionine and protein N-terminal acetylation were allowed as variable modifications and all cysteines were treated

as being carbamidomethylated. Peptide matches were filtered using false discovery rates (FDR) for peptide spectrum matches and proteins of 2% and 1% respectively.

Data analysis: Log2 transformed Label Free Quantitation (LFQ) or intensity Based Absolute Quantitation (iBAQ) values (*Schwanhäusser et al., 2011*) were used for the analysis. To assess loading across the six conditions, each as biological duplicate with technical triplicates (1st sample) or technical duplicates (2nd sample), three metabolic enzymes (Enolase 2, Galactokinase and Glyceraldehyde-3-phosphate dehydrogenase), which we considered to be 'innocent bystanders' were examined. The signal for the three proteins were comparable between all samples. Hereafter we used the MS2-protein signal to adjust the iBAQ values for each ribosome biogenesis protein of interest. Data are available in *Supplementary file 1*. Data were processed using Perseus v 1.6.0.7 (*Tyanova et al., 2016*).

## Data availability

The cryo-EM density map for the 5′ ETS RNP has been deposited in the EM Data Bank with accession code EMD-0441. Coordinates for the 5′ ETS RNP have been deposited in the Protein Data Bank under accession code 6ND4.

## Acknowledgements

We thank M Chaker-Margot and A Antar for help with the molecular cloning of constructs, M Ebrahim and J Sotiris for their exceptional support with data collection at the Evelyn Gruss Lipper Cryo-EM resource center and S Darst for critical reading of this manuscript. JB is supported by a Swiss National Science Foundation fellowship (155515). SK is supported by the Robertson Foundation, the Alfred P. Sloan Foundation, the Irma T Hirschl Trust, the Alexandrine and Alexander L Sinsheimer Fund and the NIH New Innovator Award (1DP2GM123459). The Rockefeller University Proteomics Resource Center acknowledges funding from the Leona M and Harry B Helmsley Charitable Trust and Sohn Conferences Foundation for mass spectrometer instrumentation.

## Additional information

### Funding

| Funder | Grant reference number | Author |
|---|---|---|
| Swiss National Science Foundation | 155515 | Jonas Barandun |
| National Institutes of Health | 1DP2GM123459 | Sebastian Klinge |
| Robertson Foundation | | Sebastian Klinge |
| Alfred P. Sloan Foundation | | Sebastian Klinge |
| Irma T. Hirschl Trust | | Sebastian Klinge |
| Alexandrine and Alexander L Sinsheimer Fund | | Sebastian Klinge |

The funders had no role in study design, data collection and interpretation, or the decision to submit the work for publication.

### Author contributions

Mirjam Hunziker, Conceptualization, Data curation, Formal analysis, Investigation, Methodology, Writing—review and editing; Jonas Barandun, Data curation, Formal analysis, Validation, Investigation, Visualization, Methodology, Writing—review and editing; Olga Buzovetsky, Data curation, Formal analysis, Validation, Investigation, Visualization, Writing—review and editing; Caitlin Steckler, Henrik Molina, Data curation, Formal analysis, Validation, Visualization, Methodology; Sebastian Klinge, Conceptualization, Supervision, Funding acquisition, Writing—original draft, Project administration, Writing—review and editing

## Author ORCIDs

Mirjam Hunziker (iD) https://orcid.org/0000-0002-2912-4993
Jonas Barandun (iD) https://orcid.org/0000-0003-2971-8190
Sebastian Klinge (iD) https://orcid.org/0000-0002-9373-4737

## Decision letter and Author response

Decision letter https://doi.org/10.7554/eLife.45185.026
Author response https://doi.org/10.7554/eLife.45185.027

## Additional files

### Supplementary files

• Supplementary file 1. Mass spectrometry analysis of pre-ribosomal particles.
DOI: https://doi.org/10.7554/eLife.45185.017

• Supplementary file 2. Proteins identified by mass spectrometry analysis of purified 5' ETS, 5' ETS-18S$^{5'\text{-domain}}$, 5' ETS-18S$^{central\text{-domain}}$ particles used for structural studies shown in *Figure 1—figure supplement 2*. Proteins associated with the 5' ETS (yellow), 5' domain (pink), central domain (light blue) are highlighted in bold.
DOI: https://doi.org/10.7554/eLife.45185.018

• Supplementary file 3. Primers, Northern blotting probes, yeast strains and plasmids used in this study.
DOI: https://doi.org/10.7554/eLife.45185.019

• Transparent reporting form
DOI: https://doi.org/10.7554/eLife.45185.020

### Data availability

The cryo-EM density map for the 5' ETS particle has been deposited in the EM Data Bank with accession code EMD-0441. Coordinates for the 5' ETS particle have been deposited in the Protein Data Bank under accession code 6ND4.

The following datasets were generated:

| Author(s) | Year | Dataset title | Dataset URL | Database and Identifier |
|---|---|---|---|---|
| Hunziker M, Barandun J, Buzovetsky O, Steckler C, Molina H, Klinge S | 2019 | Coordinates for the 5' ETS particle | http://www.rcsb.org/structure/6ND4 | Protein Data Bank, 6ND4 |
| Hunziker M, Barandun J, Buzovetsky O, Steckler C, Molina H, Klinge S | 2019 | Cryo-EM density map for the 5' ETS particle | https://www.ebi.ac.uk/pdbe/entry/emdb/EMD-0441 | Electron Microscopy Data Bank, EMD-0441 |

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
