## [Decision Letter]

Thank you for submitting your article "Conformational switches control early maturation of the eukaryotic small ribosomal subunit" for consideration by *eLife*. Your article has been reviewed by four peer reviewers and the evaluation has been overseen by Cynthia Wolberger as the Senior Editor. The following individuals involved in review of your submission have agreed to reveal their identity: John Woolford (Reviewer #3), Katrin Karbstein (Reviewer #4).

The reviewers have discussed the reviews with one another and the Reviewing Editor has drafted this decision to help you prepare a revised submission.

I want to stress that all reviewers largely agree on the strengths and shortcomings of the manuscript, and that their comments and concerns are highly overlapping. I hope that you find these comments constructive and helpful in revising this work.

Summary:

Conformational changes that accompany the earliest stages of ribosome biogenesis are crucial for setting up assembly and maturation of the particles, yet are difficult to observe because the earliest intermediates are short lived. The authors of this manuscript used a clever approach to address this problem by expressing fragments of the pre-rRNA in yeast, which cannot mature. Comparison of the proteins bound to these rRNA fragments (by MS), and the structures (by cryo-EM) of partial assemblies with the contents and structure of the processome (the downstream intermediate) suggest that (i) individual domains can assemble proteins bound to them independently of other domains, that (ii) assembly of the full processome requires the presence of all domains (iii) that UtpB undergoes a major opening during assembly to support the recruitment of late binding factors. Overall, this manuscript is considered of high interest to the field, but major improvements in the presentation and explicit mentioning of the caveats of the system must be provided. In addition, there are some requests for additional experimental data.

Essential revisions:

Necessary experimental revisions:

1) Please analyze the binding of proteins to the 3' domain. In order to make the argument that each domain can bind its interaction partners directly, it was felt that this piece of data was needed.

2) Please provide a biological replicate of the mass spectrometry data, as well as a more detailed description of what is meant by "technical" replicates. In addition, stoichiometry of the assembly factors (relative to rRNA or r-proteins) must be addressed.

3) Please map the cleavage products in Figure 4—figure supplement 2 to support or refute the claim that these molecules are biologically relevant. (The 5' ETS probe and the A0-A1 show RNA pieces of the length expected for just A1 (no A0) processing for the RNPs containing ending after the 5', central and 3' major domains. For the one ending at the 3' minor domain, there is a piece consistent with A0 cleavage. Further supporting the notion that this is the 5' ETS ending at A0, this latter piece disappears with the A0-A1 probe.) This figure demonstrates A1 cleavage in all molecules but A0 cleavage only after the SSU processome is assembled (and is directly contradicting statements by the authors). This could be reconciled if A1 cleavage is actually cleavage near A1.

Necessary revisions in text:

1) Caveats must be introduced into the manuscript in multiple instances:

a) Whether or not these molecules are functionally relevant cannot be ascertained, and this must be clearly acknowledged. The surprising lower stringency of A1 cleavage relative to A0 cleavage must also be clearly acknowledged as suggesting that these intermediates differ somehow from real physiological intermediates.

b) Whether a trans-mechanism for assembly (an endogenous 5' ETS binding to the independent domains to facilitate assembly) could be functional as suggested by a reviewer should be acknowledged or ruled out in the ts PolI strains from Nomura.

2) The existing literature must be discussed more thoroughly. This includes work on the assembly of individual domains, as well as changes during assembly visualized by other methods. Furthermore, the literature should be inspected for evidence that some of the factors recruited late are recruited in response to a conformational change (change in salt stability etc.).

3) The authors should analyze the "class 3" particles or address why they excluded these from their analysis. This is especially important because an eye-ball analysis suggests it might contain UtpB in the "mature" conformation.

4) The authors must revise their description of the molecule not being cleaved, taking account of the actual Northern data in Figure 4—figure supplement 2.

5) The authors must better describe that the "5' ETS particle" is and if it contains more than just the 5' ETS, as believed, its name must be revised.

6) Figure organization and clarity should be improved. Specific comments include:

a) For Figure 1, I would suggest turning the RNA around (from yellow to black), as we are used to read from top to bottom. The same must happen for Figure 1B, and in panel B I would also suggest that the authors include the data from their previous work on the 5' ETS, and then go in the following order 5' ETS, 5' domain, central domain, etc. This would be the order of assembly from right to left. In Figure 1C, I think the authors should have the structures from Figure 1—figure supplement 2A-C, aligned with the rest of the figure.

b) Several reviewers comment on Figure 2 being very busy. I would suggest having the 5' ETS left and the SSU processome right, and then keep that throughout the figure. It was suggested to zoom in independently to each of the late factor binding sites. E.g., in Figure 2B it is unclear where Rrp5 is, and how that changes. Nop14 is not in the figure but referred to. Changes in the Noc4 binding site are unclear. Critically, in Figure 2C it is not clearly visible that A0 and A1 are uncut. That cleavage site needs to be pointed out.

c) The data currently presented in Supplementary Figure 6 need to be in the main paper. This is a major conclusion, perhaps *the* major conclusion of the work. I would also encourage the authors to turn the presentation in the manuscript around: it is closed in the 5' ETS RNP and opens up. This is (they think) the order in which assembly happens.

d) The legend for Figure 4 needs to stress that these are inferred not observed. It should be added where these inferences are coming from (which is totally unclear to me), and it needs to be clarified in the legend that some AFs leave the complex according to this model.

e) Figure 4—figure supplement 1C: Can the authors label the protein bands in the gels?

[Editors' note: further revisions were requested prior to acceptance, as described below.]

Thank you for resubmitting your work entitled "Conformational switches control early maturation of the eukaryotic small ribosomal subunit" for further consideration at *eLife*. Your revised article has been favorably evaluated by Cynthia Wolberger (Senior Editor), a Reviewing Editor, and two reviewers.

The manuscript has been improved but there are some remaining issues that need to be addressed before acceptance, as outlined below:

There is no concern that you were unable to assemble just the 3' domain. But there is unanimous agreement that this requires you to qualify some of your statements, including in the Abstract, which currently states: "Here we show that *each* rRNA domain of the small ribosomal subunit can independently recruit its own biogenesis factors in vivo." Similar statements are found in the Introduction and Discussion.

All reviewers are also unanimous that a qualifying sentence about the functionality would be "scholarly", given that you cannot show it due to the limitations of the assay. This could be along the lines of: "While we cannot rule out the possibility that these are off-pathway intermediates, the following strongly suggests that this is not the case." You do not need to use that wording, but you get the idea. I also want to make it clear that all reviewers specifically state that they do not feel that this affects the impact of the work.

There is also unanimous agreement that the wording 5' ETS particle is inappropriate. In the 2015 paper that you reference you actually did use a 5' ETS RNA and found 26 proteins assemble. Nonetheless in here you are studying the structure of an RNP containing 5' ETS, and the 5' and central domains of 18S rRNA. Thus, you should designate the molecule accordingly. I think the confusion comes from the fact that you refer to 5' ETS -particle as a substructure, while the writing as well as the wording particle implies a molecule.

Examples include: "We determined the cryo-EM structures of the 5' ETS (delete particle) in the context of […]", then later in the same paragraph: "[…], we observed the 5' ETS substructure/RNP as the unifying and stable structural entity […]."

"Strikingly the 5' ETS substructure/ RNP etc… adopts a structure […]".

I am sure there are more. All four reviewers, all experts in the field, did not understand which molecule you are studying, or that you refer to a changing substructure. That suggests the writing is unclear.

Finally, in Figure 1, it's currently impossible to distinguish between missing values and measurements of low abundance as they are both white – this should be changed so that the figure can be interpreted.

---

## [Author Response]

Essential revisions:Necessary experimental revisions:1) Please analyze the binding of proteins to the 3' domain. In order to make the argument that each domain can bind its interaction partners directly, it was felt that this piece of data was needed.

We thank the reviewers for this suggestion. Similar to the other domains (5’ and central domain), we have expressed the 3’ major domain with MS2 aptamers at the 3’ end. However, in contrast to the 5’ and central domain, this construct in isolation is not stable when expressed in yeast as northern blotting of total cellular RNA showed that only the MS2 aptamers remain in yeast cells (Figure 4—figure supplement 1). The degradation of this construct in vivotherefore prohibits the analysis of associated proteins. To clarify this point, we also added the following statement to the Results (subsection “18S rRNA domains independently recruit their assembly factors”): *“*While the 5’ domain and central domain could be stably expressed in isolation, the 3’ major domain was unstable when expressed individually, which prevented a direct analysis of bound factors (Figure 1B, Figure 4—figure supplement 1A,B).*”*

2) Please provide a biological replicate of the mass spectrometry data, as well as a more detailed description of what is meant by "technical" replicates. In addition, stoichiometry of the assembly factors (relative to rRNA or r-proteins) must be addressed.

For biological replicates, we have re-expressed all rRNA constructs used for Figure 1B in yeast. Subsequently we have performed two independent RNA-protein purifications for mass spectrometry analysis for each of these constructs, which are technical replicates as they originate from the same biological source. We have combined all the data in the revised version of Figure 1B, which now contains mass spectrometry data for each construct in quintuplets; three technical replicates of the first biological duplicate (as submitted originally) and two technical replicates of the second biological duplicate (as requested by the reviewers).

We note that the determination of an accurate stoichiometry between ribosomal proteins, ribosome assembly factors and RNAs is not possible with this approach. This is evidenced by the fact that any attempts of determining accurate stoichiometries in previous studies using this method (Chaker-Margot et al., 2015, Zhang et al., 2016, Chen et al., 2017, Chaker-Margot and Klinge, 2019) have been unsuccessful when comparing these data to subsequent structural data on the small subunit processome (Barandun et al., 2017) and large subunit precursors (Kater et al., 2017, Sanghai et al., 2018, Zhou et al., 2018).

3) Please map the cleavage products in Figure 4—figure supplement 2 to support or refute the claim that these molecules are biologically relevant. (The 5' ETS probe and the A0-A1 show RNA pieces of the length expected for just A1 (no A0) processing for the RNPs containing ending after the 5', central and 3' major domains. For the one ending at the 3' minor domain, there is a piece consistent with A0 cleavage. Further supporting the notion that this is the 5' ETS ending at A0, this latter piece disappears with the A0-A1 probe.) This figure demonstrates A1 cleavage in all molecules but A0 cleavage only after the SSU processome is assembled (and is directly contradicting statements by the authors). This could be reconciled if A1 cleavage is actually cleavage near A1.

In our original submission a northern blot with an MS2 probe of total cellular RNA highlighted that in vivoeach of the constructs was present with the expected length as the dominant species (Figure 4—figure supplement 2B). We also showed that specifically upon purification of these particles, degradation products appear in the northern blots when the MS2, 5’ ETS or A0-A1 probes are used (Figure 4—figure supplement 2C-E). This suggested that degradation occurs during the purification of these particles. The size of the major degradation product above 750 nucleotides further suggested non-specific cleavage events since a site-specific cleavage at site A1 would generate a product of exactly 700 nucleotides for the 5’ ETS and A0-A1 probes.

To further exclude the possibility that A1 cleavage occurs within these constructs, we have used the construct terminating with the central domain (5’ ETS-18S^central domain^), which showed the most prominent degradation product with the 5’ ETS probe (Figure 4—figure supplement 2D).

In a new supplementary figure (Figure 4—figure supplement 3) we have probed the representative construct (5’ ETS-18S^central domain^) with the previous probes (MS2, 5’ ETS and A0-A1) as well as a probe that binds immediately downstream of the A1 cleavage site (A1 probe) within the 5’ domain. As shown in the new supplementary figure (Figure 4—figure supplement 3), the major degradation product, which runs above 750 nucleotides is detected by the 5’ ETS, A0-A1 and A1 probes. This proves two important points:

1) Non-specific cleavage, and not A1 cleavage, is responsible for the observed degradation product. This most likely occurs during the purification.

2) The pre-ribosomal particles generated by our method are therefore biologically relevant since A1 cleavage would not be expected.

These data are further consistent with our model since early particles that are not competent to form the SSU processome do not undergo A0 or A1 cleavage whereas A0 cleavage is required for SSU processome formation to occur. First, our data conclusively rule out A1 cleavage for any of the incomplete pre-ribosomal RNA mimics (5’ ETS-MS2x5; 5’ ETS-18S^5’ domain^; 5’ ETS-18S^central domain^; 5’ETS-18S^3’ major domain^). Second, for the onlyconstruct that can form a small subunit processome (5’ETS-18S^3’ minor domain^) we observe A0 cleavage (with a fragment of approximately 600 nucleotides; much smaller than the degradation product), which is expected based on our previous detailed characterization of the pre-ribosomal RNAs present in the small subunit processome (Chaker-Margot et al., 2017, Figure S11A-B).

Necessary revisions in text:1) Caveats must be introduced into the manuscript in multiple instances:a) Whether or not these molecules are functionally relevant cannot be ascertained, and this must be clearly acknowledged. The surprising lower stringency of A1 cleavage relative to A0 cleavage must also be clearly acknowledged as suggesting that these intermediates differ somehow from real physiological intermediates.

We believe that the new supplementary figure (Figure 4—figure supplement 3) has addressed the concerns raised in point 3 where additional experiments were requested. In summary, the additional experiments showed that no A1 cleavage occurs in these constructs until the SSU processome is formed. We therefore believe that these preribosomal RNAs are indeed functionally relevant.

b) Whether a trans-mechanism for assembly (an endogenous 5' ETS binding to the independent domains to facilitate assembly) could be functional as suggested by a reviewer should be acknowledged or ruled out in the ts PolI strains from Nomura.

We believe that a trans-mechanism is highly unlikely for several reasons.

First, small subunit processome formation is an intramolecular reaction as far as preribosomal RNA and its associated factors are concerned.

Second, within the small subunit processome the proximity of additional assembly factors (Sas10, Mpp10, Utp11 to name just a few) and U3 snoRNA, which have multiple interactions with different domains of the same pre-ribosomal RNA, further favor an intramolecular reaction as their high local concentrations will easily outcompete assembly factors that could interact in trans.

Third, if there was a trans-mechanism, an evolutionary separation of the 5’ ETS and the 18S rRNA coding region would be expected as each sequence could exist in isolation. However, instead the 5’ ETS always directly precedes the 18S rRNA coding region.

Fourth, if a 5’ ETS particle could assemble other ribosomal RNA domains in trans, we would expect to have seen mass spectrometry evidence of these interactions in the form of 5’ ETS associated factors in the pulldowns for just the 5’ or central domains. However, this is not the case (Figure 1A, B).

In light of the four points outlined above we do not think it is appropriate to mention a trans-mechanism in this manuscript.

2) The existing literature must be discussed more thoroughly. This includes work on the assembly of individual domains, as well as changes during assembly visualized by other methods. Furthermore, the literature should be inspected for evidence that some of the factors recruited late are recruited in response to a conformational change (change in salt stability etc.).

We thank the reviewers for this comment and have expanded the Introduction to include prior literature on individual domains and their associated factors. In particular we have highlighted the use of Miller spreads, which we believe is highly relevant for this manuscript and the proposed model.

Regarding assembly factor recruitment in response to a conformational change, previous work from our laboratory as well as that of Keqiong Ye’s group has shown that the addition of 167 nt of the 3’ minor domain is sufficient for the recruitment of a large number of factors to small subunit processomes (Chaker-Margot et al., 2015; Zhang et al., 2016). Both laboratories postulated that the addition of the 3’ minor domain is not a direct binding site for these factors and this was further confirmed by all structures of small subunit processomes showing that these factors bind on the outside of the small subunit processome and not the 3' minor domain directly (Kornprobst et al., 2016; Chaker-Margot et al., 2017; Sun et al., 2017; Barandun et al., 2017; Cheng et al., 2017). This observation can only be rationalized by the creation of new binding sites for these factors, which previously did not exist as the individual domains of the SSU processome (5’ ETS, 5’ domain, central domain, 3’ major domain and 3’ minor domain) were physically separated from one another. The structural changes of UtpB first described in this manuscript lend additional support to this model.

3) The authors should analyze the "class 3" particles or address why they excluded these from their analysis. This is especially important because an eye-ball analysis suggests it might contain UtpB in the "mature" conformation.

To analyze the state of UtpB within class 3, this class was subjected to a subsequent 3D classification that revealed that only 22% (less than 5% of the entire dataset) contain the isolated tetramerization domain of UtpB in a mature configuration. The small number of particles and limiting resolution of the resulting reconstruction prevented further analysis. Since 64% of all particles contain the compacted configuration of UtpB, which we also observe in the reconstructions obtained for just the 5’ ETS as well as the 5’ ETS -5’ domain construct, we believe that this dominant state is physiologically relevant.

4) The authors must revise their description of the molecule not being cleaved, taking account of the actual Northern data in Figure 4—figure supplement 2.

The actual Northern data (now presented in Figure 4—figure supplement 2 and 3) show that cleavage at site A1 has not occurred.

5) The authors must better describe that the "5' ETS particle" is, and if it contains more than just the 5' ETS, as believed, its name must be revised.

Originally, we coined the term “5’ ETS particle” for a particle that contains the 5’ external transcribed spacer and its associated 26 proteins (Chaker-Margot et al., 2015). Since our visualized 3D reconstruction contains precisely these components, we believe that the term is appropriate. To clarify this point in the manuscript, we have added a sentence in the Introduction, which states:

“The particle containing the 5’ ETS region and its associated 26 unique polypeptides was termed the 5’ ETS particle (Chaker-Margot et al., 2015).”

6) Figure organization and clarity should be improved. Specific comments include:a) For Figure 1, I would suggest turning the RNA around (from yellow to black), as we are used to read from top to bottom. The same must happen for Figure 1B, and in panel B I would also suggest that the authors include the data from their previous work on the 5' ETS, and then go in the following order 5' ETS, 5' domain, central domain, etc. This would be the order of assembly from right to left. In Figure 1C, I think the authors should have the structures from Figure 1—figure supplement 2A-C, aligned with the rest of the figure.

While we thank the reviewer for this suggestion, we strongly believe that Figure 1 should remain exactly as it is for the following reasons:

1) Turning panels A and B would not scientifically change the interpretation of this data.

2) We believe that including data from previously published primary literature in a new publication is problematic.

3) As only the highest resolution reconstruction is discussed in this manuscript, including other reconstructions would not help in clarifying the content of this manuscript for the readership.

b) Several reviewers comment on Figure 2 being very busy. I would suggest having the 5' ETS left and the SSU processome right, and then keep that throughout the figure. It was suggested to zoom in independently to each of the late factor binding sites. E.g., in Figure 2B it is unclear where Rrp5 is, and how that changes. Nop14 is not in the figure but referred to. Changes in the Noc4 binding site are unclear. Critically, in Figure 2C it is not clearly visible that A0 and A1 are uncut. That cleavage site needs to be pointed out.

We thank the reviewers for this suggestion and agree that Figure 2 in its previous version was quite complex. To simplify this figure (now Figure 3; see response to point 6c below), we have split it into two parts. The first part (now Figure 3) describes the consequences of conformational changes within the SSU processome and 5’ ETS particle whereas the second part (now Figure 4) describes the differences in RNA between the two particles.

To clarify the depiction of assembly factor binding sites on UtpB, we have included a visualization that shows these factors as cartoon representations on the surfaces of UtpB (Figure 3C, top part) whereas the bottom part of the same panel shows the same structures with the mapped binding sites that are labelled in italics for each of the assembly factors. Nop14 is shown in Figure 3A.

Changes in the Noc4 binding site have been re-rendered for clarification in Figure 3E and F.

To increase clarity, grey boxes have been removed and the outline width of the rendered structures has been increased.

In the RNA description (now Figure 4), sites A0 and A1 have been more clearly indicated as requested.

*c) The data currently presented in Supplementary Figure 6 need to be in the main paper. This is a major conclusion, perhaps* the *major conclusion of the work. I would also encourage the authors to turn the presentation in the manuscript around: it is closed in the 5' ETS RNP and opens up. This is (they think) the order in which assembly happens.*

We thank the reviewer for this suggestion. We have accordingly shifted the previous Supplementary Figure 6 into the main text (now Figure 2) to put additional emphasis on this point.

d) The legend for Figure 4 needs to stress that these are inferred not observed. It should be added where these inferences are coming from (which is totally unclear to me), and it needs to be clarified in the legend that some AFs leave the complex according to this model.

We have revised the figure and expanded the figure legend of what is now Figure 6 to clarify the above points.

e) Figure 4—figure supplement 1C: Can the authors label the protein bands in the gels?

Since this manuscript discusses more than 50 assembly factors, the sheer number of these proteins does not allow us to assign bands in each gel, which is why we have used in solution mass spectrometry instead.

[Editors' note: further revisions were requested prior to acceptance, as described below.]

*There is no concern that you were unable to assemble just the 3' domain. But there is unanimous agreement that this requires you to qualify some of your statements, including in the Abstract, which currently states: "Here we show that* each *rRNA domain of the small ribosomal subunit can independently recruit its own biogenesis factors in vivo." Similar statements are found in the Introduction and Discussion.*

All reviewers are also unanimous that a qualifying sentence about the functionality would be "scholarly", given that you cannot show it due to the limitations of the assay. This could be along the lines of: "While we cannot rule out the possibility that these are off-pathway intermediates, the following strongly suggests that this is not the case." You do not need to use that wording, but you get the idea. I also want to make it clear that all reviewers specifically state that they do not feel that this affects the impact of the work.There is also unanimous agreement that the wording 5' ETS particle is inappropriate. In the 2015 paper that you reference you actually did use a 5' ETS RNA and found 26 proteins assemble. Nonetheless in here you are studying the structure of an RNP containing 5' ETS, and the 5- and central domains of 18S rRNA. Thus, you should designate the molecule accordingly. I think the confusion comes from the fact that you refer to 5' ETS particle as a substructure, while the writing as well as the wording particle implies a molecule.Examples include: "We determined the cryo-EM structures of the 5' ETS (delete particle) in the context of […]", then later in the same paragraph: "[…], we observed the 5' ETS substructure/RNP as the unifying and stable structural entity […].""Strikingly the 5' ETS substructure/ RNP etc… adopts a structure […]".I am sure there are more. All four reviewers, all experts in the field, did not understand which molecule you are studying, or that you refer to a changing substructure. That suggests the writing is unclear.Finally, in Figure 1, it's currently impossible to distinguish between missing values and measurements of low abundance as they are both white – this should be changed so that the figure can be interpreted.

We have addressed the last concerns as outlined below.

1) We have specified in the manuscript (Abstract, Introduction and Discussion) that *stable* rRNA domains, in other words the 5’ and central domain of the small subunit rRNA but not the 3’ major domain, can recruit their own assembly factors.

2) By using the provided template, we have made the following statement regarding the functionality of the studied particles:

“While we cannot categorically rule out the possibility that the particles isolated in this study are off-pathway intermediates, the following observations suggest that this is not the case. Our previous work demonstrated that pre-rRNA mimics containing MS2 loops that are transcribed by RNA polymerase II can serve as the sole source of ribosomal RNA. Furthermore, in good agreement with prior genetic and biochemical data, we detected all expected ribosome assembly factors that were associated with the studied pre-rRNA mimics. Lastly, the specific A0 cleavage observed upon SSU processome formation indicates that the studied particles undergo specific quality control as expected for native particles.”

3) Thank you for the suggesting “5’ ETS RNP”. We have used this term in all instances where previous terminology was ambiguous.

4) We have revised Figure 1 such that proteins that were not detected are now highlighted in light brown.